# The Analysis of Trends in Survival for Patients with Melanoma Brain Metastases with Introduction of Novel Therapeutic Options before the Era of Combined Immunotherapy—Multicenter Italian–Polish Report

**DOI:** 10.3390/cancers14235763

**Published:** 2022-11-23

**Authors:** Joanna Placzke, Paweł Teterycz, Pietro Quaglino, Bozena Cybulska-Stopa, Marco Tucci, Marco Rubatto, Tomasz Skora, Valeria Interno, Magdalena Rosinska, Aneta Borkowska, Anna Szumera-Cieckiewicz, Mario Mandala, Piotr Rutkowski

**Affiliations:** 1Department of Soft Tissue/Bone Sarcoma and Melanoma, Maria Sklodowska-Curie National Research Institute of Oncology, ul. Roentgena 5, 02-781 Warsaw, Poland; 2Department of Computational Oncology, Maria Sklodowska-Curie National Research Institute of Oncology, ul. Roentgena 5, 02-781 Warsaw, Poland; 3Department of Medical Sciences, Dermatology Clinic, University of Turin, 10124 Turin, Italy; 4Department of Clinical Oncology, Maria Sklodowska-Curie National Research Institute of Oncology, 31-115 Kraków, Poland; 5Oncogenomic and Innovative Therapies and Medical Oncology Unit, Department of Interdisciplinary Medicine, University of Bari ‘Aldo Moro’, Piazza Giulio Cesare 11, 70124 Bari, Italy; 6Medical Oncology Unit, Department of Interdisciplinary Medicine, University of Bari ‘Aldo Moro’, Piazza Giulio Cesare 11, 70124 Bari, Italy; 7Department of Pathology, Maria Sklodowska-Curie National Research Institute of Oncology, ul. Roentgena 5, 02-781 Warsaw, Poland; 8Unit of Medical Oncology, Department of Surgery and Medicine, University of Perugia, Santa Maria Misericordia Hospital, 06129 Perugia, Italy

**Keywords:** melanoma brain metastases, brain metastases treatment, symptoms, steroids, mol-GPA, prognostic index

## Abstract

**Simple Summary:**

The landscape of treatment of patients with melanoma brain metastases (MBM) is continually evolving. We report the real-world data on 531 internationally treated patients with MBM before and after 2017 and their prognosis and treatment outcomes before the introduction of combined immunotherapy. We aimed to analyze trends in survival probability and their relevance to the currently used prognostic index, melanoma mol-GPA, and to some presumed prognostic and predictive factors not included in mol-GPA, such as symptoms occurrence and use of steroids. We have observed significant improvement in the survival of patients with the poorest mol-GPA prognosis. In our prognostic model, the presence of symptoms associated with brain metastases predicted a worse response to immune checkpoint inhibitors; however, symptoms without steroid use did not have prognostic significance. The prognosis of patients with MBM has been improving over the years due to the introduction of modern local and systemic treatment options across all mol-GPA prognostic groups.

**Abstract:**

Stage IV melanoma patients develop melanoma brain metastases (MBM) in 50% of cases. Their prognosis is improving, and its understanding outside the context of clinical trials is relevant. We have retrospectively analyzed the clinical data, course of treatment, and outcomes of 531 subsequent stage IV melanoma patients with BM treated in five reference Italian and Polish melanoma centers between 2014 and 2021. Patients with MBM after 2017 had a better prognosis, with a significantly improved median of overall survival (OS) after 2017 in the worst mol-GPA prognostic groups (mol-GPA ≤ 2): a median OS >6 months and HR 0.76 vs. those treated before 2017 (CI: 0.60–0.97, *p* = 0.027). In our prognostic model, mol-GPA was highly predictive for survival, and symptoms without steroid use did not have prognostic significance. Local therapy significantly improved survival regardless of the year of diagnosis (treated before or after 2017), with median survival >12 months. Systemic therapy improved outcomes when it was combined with local therapy. Local surgery was associated with improved OS regardless of the timing related to treatment start (i.e., before or after 30 days from MBM diagnosis). Local and systemic treatment significantly prolong survival for the poorest mol-GPA prognosis. Use of modern treatment modalities is justified in all mol-GPA prognostic groups.

## 1. Introduction

The most common cause of death among melanoma patients is the presence of brain metastases, which develop in 50% of patients with stage IV disease. Around 30% of patients with metastatic stage melanoma present with melanoma brain metastases (MBM) at diagnosis [1]. The median survival of patients with MBM was 4–5 months before the novel therapeutic methods were introduced [1]. The progression-free and overall survival of patients with MBM improved substantially with the new treatment modalities application. Patients with melanoma brain metastases respond to immune checkpoint inhibitors (ICI) and BRAF/MEK inhibitors (BRAF/MEKi); however, they are underrepresented in clinical trials [2,3,4]. Understanding the prognostic impact of new therapeutic options outside the context of clinical trials is less clear for patients with intracranial than with extracranial dissemination [2]. Current treatment options include surgery, whole brain radiation (WBRT), stereotactic radiation (SRS), systemic therapy (BRAF/MEKi, ICI, chemotherapy), and best supportive care. In 2016, two main treatment strategies for melanoma patients, ICI and BRAF/MEKi, were registered in Europe. Since 2017, these systemic therapies have been given concomitantly or in sequence with surgery or stereotactic radiotherapy (SRS) and have become the standard of care in melanoma patients with brain metastases [2,3].

Various factors influencing patients’ prognosis should be considered before making therapeutic decisions in patients with MBM. The prognosis of patients with brain metastases can be estimated and depends on many factors [1,5,6,7,8,9]. Among them are histology and molecular profile of a tumor. Organ-specific factors that have significant influence on prognosis were grouped into prognostic indices, or graded scales, such as Diagnosis-Specific Graded Prognostic Assessment (DS-GPA) [10]. Unique prognostic models were developed for patients with non-small cell lung cancer, small cell lung cancer, melanoma, renal cell carcinoma, breast cancer, and gastrointestinal cancer. Melanoma mol-GPA (molecular GPA; Graded Prognostic Assessment) is composed of five factors: age, Karnofsky Performance Score (KPS), presence of extracranial metastases, number of brain metastases, and *BRAF* mutation status. Factors are assigned a value of 0, 0.5, or 1.0. From these data, the GPA score is calculated and the specific survival is predicted. The best prognosis has mol-GPA group scoring from 3.5 to 4.0 points with OS reaching 34 months, while in the mol-GPA 0–1-point group with the worst prognosis OS is only 5 months. GPA is designed to distinguish classes of patients by prognosis before treatment [3,4,11,12]. Included in melanoma mol-GPA index, *BRAF* mutational status is closely related to the more aggressive behavior of melanoma in a metastatic setting and is a predictor for the choice of therapy. It is unclear if the prognosis affects the treatment sensitivity (outcomes) and if melanoma mol-GPA index should guide treatment decisions [13]. We aspired to analyze our past treatment choices and navigate future ones. We evaluated them in relation to the available results of prospective clinical trials dedicated to patients with melanoma brain metastases: Checkmate 204, Australian ABC, and COMBI-MB, which mostly included asymptomatic patients and demonstrated that those patients benefit the most from the modern treatment approach of combined immunotherapy or combined BRAF/MEK inhibitor therapy [11,12,13]. The role of single-agent ICI in MBM treatment was evaluated in three prospectively performed clinical trials. In asymptomatic patients, the intracranial objective response rate (ORR) was 16% for ipilimumab [14], 26% for pembrolizumab [15], and 20% for nivolumab [12]. Symptomatic patients with MBM were underrepresented in those trials. They are difficult to treat, and it is unclear what treatment options should be used. In symptomatic patients with MBM, the response rates to therapy are much worse. ICI monotherapy provides intracranial response rates for symptomatic pts on the level of 5% and 6% for ipilimumab and nivolumab, respectively, and on the level of 59% for BRAF/MEKi therapy but with a median of 4.5 month response duration [13,14,16]. Careful evaluation of symptomatic patients with MBM is critical because of the different reasons for symptom occurrence (mass effect, involvement of critical structures, edema, or midline shift) [11]. Symptoms may be associated with disease localisation in the brain or rapid disease progression and frequently require local therapy and/or short- or long-term glucocorticoid use to control neurological complications or deficits. We analyzed the outcomes of MBM patients’ treatment limited to local and systemic therapy before the introduction of combined immunotherapy to the routine practice. We wanted to analyze the trends in survival probability in the real-world dataset of internationally treated patients with melanoma brain metastases and their relevance to the currently used prognostic index, melanoma mol-GPA, and to some presumed prognostic and predictive factors not included in mol-GPA, such as symptoms occurrence and use of steroids.

## 2. Materials and Methods

### 2.1. Patient Cohort and Inclusion Criteria

In this international, multicenter, retrospective study, we analyzed baseline characteristics, course of treatment, and clinical outcomes of all consecutive patients with newly diagnosed MBM who were treated in three Italian and two Polish reference cancer centers between January 2014 and March 2021.

We have compared patients diagnosed with MBM before and after 1 January 2017, as it was the year when the systemic treatment of patients with metastatic melanoma with BRAF/MEKi and ICI started to become widely used both in Italy and Poland due to the implementation of national reimbursed programs.

We included all patients eligible for at least one systemic or local treatment modality, for whom at least one radiological (contrast-enhanced CT or MRI) assessment after at least three months of treatment or information on earlier clinical progression/death was available.

### 2.2. Collected Covariates

Patients’ data were retrieved retrospectively from their medical history records. The baseline information included the date of primary diagnosis, *BRAF* V600 mutation status, date of MBM diagnosis, number of central nervous system (CNS) metastases, maximal diameter of CNS metastases, Karnofsky score at MBM diagnosis, CNS symptoms, treatment with glucocorticoids (GCs) at MBM diagnosis, and previous treatment lines. The MBM treatment data included consecutive systemic therapies, surgical interventions, and SRS and WBRT. In the follow-up data, intracranial and extracranial progression and last observation (or death) were recorded.

### 2.3. Analysis Plan and Handling of Missing Data

The analysis was split into two parts. The first deals with the importance of prognostic baseline covariates. It was performed on data after multiple imputations by chained equations (MICE) procedure, assuming missing at random data. The purpose of this part was to examine possible prognostic variables in a multivariable model and perform independent validation of the melanoma mol-GPA index that consists of 5 factors, KPS, number of MBM, *BRAF* status, age, and extracranial metastases, and divides patients into 4 prognostic groups according to the assigned scores given for factors, 0–1 scores have the worst prognosis, and 3.5–4.0 scores have the best prognosis [17]. The prognostic models were built by pooling results over 40 imputed datasets and the random forest method for both numeric and categorical variables. The details of the applied MICE procedure are presented in Appendix A.

The second part describes patterns of treatment in patients with MBM. This part was performed only on complete cases; i.e., we have decided to exclude patients with missing data regarding age, presence of extracranial metastases, number of metastases in the CNS, Karnofsky score, and *BRAF* status (variables present in mol-GPA classification) due to probable correlation with therapy selection and a high chance of introducing bias to treatment efficacy estimation through imputation procedure. In this part, we examined the therapy results. As the treatment selection is closely related to the underlying disease burden, we did we did not attempt to build a prognostic model and compare treatment efficacy in this group.

### 2.4. Statistical Methods

The continuous variables were summarized by the median and interquartile range (IQR), while categorical variables were summarized by number and percentage of total cases. Unless otherwise stated, all point estimates were reported with a 95% confidence interval (CI). All the tests were two-sided.

Overall survival (OS) was calculated from the date of MBM diagnosis up to death or last observation. Patients alive at the last observation were censored. Progression-free survival (PFS) was calculated from the start of therapy up to radiological or clinical progression or death. Intracranial PFS was calculated from the start of therapy up to radiologically confirmed progression of lesions in the CNS or death. Extracranial PFS was calculated from the start of therapy to the radiologically confirmed progression outside of CNS or death. Patients alive and without respective progression were censored at the last observation. The survival analysis was performed using the Kaplan–Meier estimator with the log-rank test and the Cox proportional hazard model.

All analyses were performed using the R language environment version 4.1.2 (The R Foundation for Statistical Computing) with abundant use of tidyverse and survminer packages [18,19]. A *p* < 0.05 was considered statistically significant. No adjustment for multiple testing was applied. Multivariable models were assessed using the concordance index and Akaike Information Criterion (AIC) [20].

### 2.5. Ethical Statement

The study was conducted according to the guidelines of the Declaration of Helsinki and was approved by the Institutional Review Board and Bioethics Committee at the Maria Sklodowska-Curie National Research Institute of Oncology; ethics board approval registration number KB/430-74/20 (“Long Term Results and Prognostic Biomarkers in Advanced Cutaneous Melanoma Patients”). All patients signed an informed consent form for treatment in accordance with standard operating procedures used in our hospitals. In addition, patients were diagnosed and treated in accordance with national guidelines and policies.

## 3. Results

### 3.1. Patient Characteristics

Electronic health records of 531 patients who were diagnosed with MBM between January 2014 and March 2021 were retrieved from five European reference cancer centers for this study. After applying the inclusion criteria, 380 pts from Poland and 151 pts from Italy were included. Figure 1 shows numbers of patients included in each portion of our analysis. Patients’ clinical characteristics are summarized in Table 1. Categorization into distinct mol-GPA score groups is shown in Appendix A.

### 3.2. Prognostic Factors at MBM Diagnosis

The median overall survival (OS) time for patients included in prognostic models was 6.8 months (CI: 5.9–7.9). In *BRAF V600* mutated and wild-type melanomas, the OS was 7.9 months (CI: 6.8–9.0) and 5.3 months (4.5–6.3), respectively. At the time of the analysis, 68 patients (14.0%) were alive. Table 2 shows two multivariate Cox models pooled from 40 multiple imputed datasets. The “full model” comprises all baseline variables of interest. In the “reduced model,” only variables with *p* < 0.1 in the full model were included. In the post hoc analysis, steroid use and CNS symptoms were highly correlated and contributed to artificial variation inflation. CNS symptoms were left in the model because of their higher clinical importance. The concordance for the “reduced model” presented in Table 2 is 0.68 (CI: 0.66–0.71), with an AIC of 4426, while the model based on mol-GPA score adjusted for the year of diagnosis had a concordance of 0.64 (CI: 0.61–0.67) and an AIC of 4470.

Mol-GPA score was highly distinctive for the prognosis regarding OS. Furthermore, we observed a difference in survival across mol-GPA groups depending on the date of CNS-involvement diagnosis. Patients diagnosed with MBM after 2017 had better prognosis. This difference was numerically observed in most of the mol-GPA score groups, and it was statistically significant in patients with the poorest prognosis (i.e., mol-GPA ≤ 2), where HR for OS reached 0.76 for patients treated after vs. before 2017 (CI: 0.60–0.97, *p* = 0.027). In patients with a better prognosis: the mol-GPA > 2, HR for OS was 0.83 (CI: 0.56–1.15. *p* = 0.26). Data on the HR for OS by the mol-GPA groups is presented in Table 3 and visualized in Figure 2. The median survival time for the worst and the best prognostic groups according to mol-GPA in our cohort improved before and after 2017, from 2.7 to 4.6 and from 15.6 to 21.1 months, respectively. The median OS by mol-GPA score groups is presented in Appendix A.

In our reduced prognostic model, the presence of MBM symptoms without glucocorticoids (GCS) use does not have prognostic significance; however, a lack of CNS symptoms with GCS usage due to other causes (e.g., preventive for asymptomatic edema, during radiotherapy or surgery) has negatively influenced prognosis. Symptomatic MBM requiring steroids had significantly worse prognosis. These findings are presented in Table 4.

We have included data on treatment before CNS dissemination diagnosis in our prognostic model, as systemic or local treatment has become more popular in various disease stages of melanoma. In our data, any line of previous systemic treatment significantly worsened the prognosis of patients with MBM.

### 3.3. Treatment

Among 402 patients with MBM included in the treatment part of the analysis, 235 patients were treated from 2014 to 2016 and 252 from 2017 to 2021. The treatment was started within 30 days from the MBM diagnosis in 293 (72.9%) cases and after 30 days from MBM diagnosis in 109 cases. The primary treatment extensively depended on clinical variables such as the number of metastases, previous therapies, and size of the largest metastatic lesions.

Among patients eligible for radiotherapy, the percentage of patients treated with SRS increased from 19% in the 2014–2016 period to 38% in 2017–2021. Similarly, in 2017–2021, the percentage of patients treated with anti-PD1 based therapy or a combination of BRAF/MEKi increased from 55% to 90%. After 2017, only two patients (1.2%) did not receive novel therapy as the first line of MBM treatment. The outcome of patients with BMs improved significantly when local therapy (neurosurgery ± SRS) was combined with modern systemic therapy, regardless of timing (*p* < 0.001). The implementation of systemic therapy and surgery was associated with improved OS before or after 30 days from MBM diagnosis. In univariable analysis, systemic treatment given within or outside 30 days from MBM diagnosis significantly reduced the risk of death from MBM with HR: 0.34 (0.26–0.45, *p* < 0.001) and HR: 0.26 (0.20–0.34, *p* < 0.001), respectively. Stereotactic radiotherapy given before or after 30 days from MBM diagnosis improved survival significantly with HR: 0.45 (0.29–0.70) and HR: 0.36 (0.24–0.55), respectively (*p* < 0.001), and the data are presented in Appendix A. Previous systemic therapy and whole brain radiotherapy (WBRT) given within 30 days from MBM diagnosis significantly negatively influenced the risk of death from MBM. WBRT performed outside the 30 days interval to MBM diagnosis did not impact OS. WBRT usage as the first treatment method in our analysis is associated with a poorer OS compared with no radiotherapy; HR: 1.42 (CI: 1.08–1.87, *p* < 0.001) and appears to depend on unfavorable patients’ characteristics. The complete comparison of treatment selection regarding the clinical status and univariable Cox models for OS, depending on treatment selection, are presented in Appendix A.

Kaplan–Meier survival curves depending on the systemic treatment used in the first line, after MBM were diagnosed, are presented in Figure 3.

The introduction of local therapy for MBM treatment significantly improved survival regardless of the year of diagnosis (treated after or before 2017), with a median survival of more than 12 months, as shown in Figure 4.

Among *BRAF*-mutated melanomas, 12 patients started PD1-based therapy and 88 started BRAF+MEK inhibitors in the first line, and their outcome correlated with the presence of symptoms associated with MBM. We have observed that the absence of symptoms significantly correlates with better overall survival only in a subgroup of patients treated with anti-PD1 antibodies. Overall survival curves of patients, who had no previous systemic therapy and started systemic therapy due to melanoma brain metastases, are presented in Figure 5.

## 4. Discussion

We would like to lead the discussion toward the most important results of this study:Patients diagnosed with MBM after 2017 had a better prognosis in our cohort, with a significantly improved median of overall survival after 2017. The biggest change was observed in the poorest prognostic mol-GPA groups, in which survival improved significantly after 2017.The outcome of patients with MBM improved when local therapy (neurosurgery ± SRS) was combined with modern systemic therapy. The implementation of systemic therapy and surgery was associated with improved OS regardless of the timing to MBM treatment start (i.e., before or after 30 days from MBM diagnosis).The introduction of local therapy for MBM treatment significantly improved survival regardless of the year of diagnosis (treated after or before 2017), with a median survival of more than 12 months.In our prognostic model, the presence of MBM symptoms without steroid use did not have prognostic significance.

We compared patients treated before and after 2017 in our analysis. Survival varied significantly by the year of diagnosis with MBM due to the introduction of new therapeutic options (ICI, BRAF/MEKi) in Europe in 2016 and their full availability in Italy and Poland in 2017, when national programs for the systemic treatment of patients with metastatic melanoma started to become widely used.

We have shown that patients diagnosed and treated for MBM in and after 2017 had significantly superior survival compared with those treated before 2017, with a clear and stable trend in rising survival probability, as seen in the literature [21,22,23,24]. Specifically, 2017 was chosen as the separator of the two treatment eras, before and after the introduction of novel therapies. We have also observed a steady and stable trend in survival improvement when a year is treated as a continuous variable due to the improving results of the patients’ treatment.

Prognostic tools, such as mol-GPA (“https://brainmetgpa.com” accessed on 30 September 2022), were developed to estimate the survival of patients with MBM regardless of the treatment. The idea of the usefulness of GPA is that it helps doctors select the appropriate treatment for their patients before its start [17]. To date, in patients with MBM, a prognostic index guided treatment choices based on prognostic factors and helped identify patients with brain metastases with a very poor prognosis, for whom only the best supportive care could be the best choice [14,17,25]. We show that mol-GPA prognosis has improved in the era of new therapies due to their usage. In our cohort, the change in prognosis over the years was the most marked and had statistical significance in the worst prognostic mol-GPA groups with a score below or equal to two (mol-GPA ≤ 2). The factor that changed survival in this group was modern treatment. Therefore, we assume that patients with the worst mol-GPA score do benefit from novel therapeutic options, as their prognosis improved significantly. The poorest survival groups estimated by the mol-GPA index do not predict the lack of treatment effectiveness in our cohort. A prognostic index is not justified to be used as a predictive one. Indeed, GPA does not have a predictive quality, and novel therapeutic approaches do conquer the biology of melanoma and can influence the prognosis.

We suggest that identifying patients with the worst prognosis according to the available indices should not limit treatment options for them, as, in our study, the median survival in the worst prognostic group has improved from 2.7 months before 2017 to 4.6 months after 2017. Some researchers even suggest escalating treatment in the worst prognostic groups while deescalating in groups of patients with very good prognosis [26].

The outcome of patients with MBM in our cohort is highly improved when local therapy (neurosurgery ± SRS) is combined with modern systemic therapy regardless of the timing related to MBM diagnosis (before or after 30 days from MBM diagnosis), being in concordance with results from other clinical trials and analyses [27,28].

In our analysis, implementation of local therapy, such as surgery or/and SRS, was associated with improved OS regardless of the year of treatment (before and after 2017). We can, therefore, assume that better prognosis after 2017 for all mol-GPA prognostic groups depends on the systemic treatment application. It cannot be ignored that the prognosis depends on the given therapy and improves over time, so, in an era of rapid change in the melanoma treatment paradigm (i.e., introduction of adjuvant and neoadjuvant therapy), the treatment factors should not be overlooked. We confirmed that the prognosis is highly dependent on the changing course of treatment over time. It seems reasonable to maximize efforts using all available treatment modalities, to improve efficacy and conquer poor prognosis. Taking advantage of the availability of all novel therapeutic options and using them is to the benefit of patients.

The melanoma mol-GPA index excludes data on symptoms, steroid usage, and treatment before CNS dissemination. These factors seem to be important while making the prognosis and predictions about MBM treatment outcomes nowadays, when there are many new early treatment options available, as in adjuvant or neoadjuvant setting. Data on symptoms, steroid usage, and previous treatment is intuitive and easy to apply. Our prognostic model supplements the acknowledged prognostic factors in the mol-GPA index. We have analyzed the prognostic value of the presence of symptomatic MBM on patients’ survival over the years. In our prognostic model, symptoms are of lesser importance provided they are steroid-free. The use of steroids in asymptomatic patients seems to be important, and it identifies patients with a worse prognosis. In our cohort, symptomatic vs. asymptomatic distinction appears to be less relevant for prognosis. We point to the significant influence of steroid use on the prediction of poorer survival of patients with MBM in real-world data, regardless of the year of diagnosis. In our cohort, the presence of symptoms associated with steroid usage in patients with melanoma brain metastases predicted worse response to immune checkpoint inhibitors. Glucocorticoids are suspected to limit the efficacy of immunotherapy, not only in melanoma patients [29]. Their influence on MBM treatment is to be estimated, hopefully soon, with the results of the ongoing clinical trial NCT03563729, which is dealing with the question of whether treatment of patients with MBM who require steroids with pembrolizumab alone is as good as with a combination of ipilimumab and nivolumab [30]. There are some indirect conclusions from prospective clinical trials devoted to patients with MBM and retrospective analysis. One of them, concerning the use of bevacizumab in patients with MBM, shows its impact on steroid use reduction in heavily pretreated patients’ population with MBM with very poor prognosis [31]. Treatment with bevacizumab led to steroid reductions and facilitated the use of immunotherapy with a durable response.

Supported by our findings, we discourage the overuse and prophylactic use of steroids during MBM treatment and encourage a quick taper whenever glucocorticoids are indispensable. According to ESMO recommendations, immunotherapy should not be administered when patients take more than 4 mg of dexamethasone a day. Immunotherapy can be used in symptomatic patients on corticosteroids at a dose < 4 mg of dexamethasone [2]. In contrast to the results from other clinical trials, in our multivariate analysis of real-world data, symptomatic patients with MBM, if steroid-free, do not have a significantly poorer prognosis, so we encourage using all available treatment modalities in their treatment. In symptomatic patients, surgery or SRS can make them asymptomatic and more prone to the effect of immunotherapy given afterwards.

Negligible numbers of symptomatic patients with MBMs are recruited into clinical trials, and, consequently, the majority of evidence report benefit observed in the asymptomatic group [12,28,32]. The CheckMate 204 trial and Australian ABC study evaluated the influence of immune checkpoint inhibitor treatment on the survival of patients with MBM. The use of steroids from the presence of steroid-free symptoms was not distinguished in those trials. Symptomatic or using steroids patients were in the same cohort. In the ABC study, symptomatic patients were in one group with those with leptomeningeal spread. It is unclear if the lack of response in symptomatic cohorts is due to a higher tumor burden and rapid growth, to a more immune-resistant phenotype, to the low distribution of tumor-infiltrating lymphocytes or to the immune suppression induced by corticosteroids [32]. Symptoms might be associated with higher disease burden or rapid disease growth deteriorating the survival prognosis. Symptoms may relate to MBM localization in the brain and, if located in eloquent regions (the left temporal and frontal lobes for speech and language, bilateral occipital lobes for vision, bilateral parietal lobes for sensation, bilateral motor cortex for movement), might not correspond to the disease burden. We would like to emphasize the distinction of the neurological symptoms at the time of systemic treatment introduction, related to the disease burden and requiring steroids from the symptoms present at diagnosis, which resolve by the treatment start or relate to the location in the brain only and thus, do not require steroids at the treatment start. The use of glucocorticosteroids, not the presence of symptoms at diagnosis, seems to worsen prognosis. According to our findings, symptoms do not interfere with prognosis. This thesis is supported by other retrospective findings, where presenting symptoms were found of no influence on prognosis [6]. If we deprive symptomatic pts of the immunotherapy, we do influence their prognosis, since some of them can achieve durable responses [6,32]. Our data point to the importance of targeting the symptomatic group of patients in the future studies.

In our analysis, the longest median overall survival was achieved by patients with MBM with the *BRAF* mutation status positive. According to our understanding, this melanoma feature impacts survival due to the possibility of having additional treatment options with BRAF/MEK inhibitors. On the other end are patients whose MBM developed on or after anti-PD1 and BRAF inhibitors and they have the poorest outcomes due to foreseen resistance to systemic therapy. Patients qualifying for local treatment have better prognosis in our analysis and the use of local therapy impacted patients’ survival regardless of the year of diagnosis and these results are comparable with those achieved by other researchers. Local therapy with stereotactic radiosurgery or surgery leads to better survival [27,28].

The strength of our analysis lies in the inclusion of a multicenter cohort of MBM patients treated both before and after the advent of novel therapies, before the era of combined immunotherapy. In this cohort, we can show directly the real-world treatment benefit of these therapies, evaluate the relevance of mol-GPA in the era of new therapies, and point to the problematic issues for making predictions about treatment outcomes and survival. We draw attention to the symptomatic group of patients with MBM, who are underrepresented in clinical trials. Our results are consistent with the trends of improved survival shown in the published literature on mol-GPA and the treatment outcomes of patients with MBM [6,17,27,33]. They are in concordance with survival estimates for control groups from recent clinical trials [11,13,16,32,34].

There are a few limitations of our analysis: the small number of patients in each treatment modality group makes it hard to perform subgroup analyses with adequate statistical power; missing data lead to the exclusion of some patients from parts of the analysis; no archived blood samples prevent exploratory analysis. For example, an interesting investigation into the possible association among mol-GPA index factors, pretreatment inflammatory peripheral blood markers, and patients’ prognosis could not be performed, due to the lack of archived samples, but can be addressed in the future in the prospective setting. Peripheral blood markers, CRP C-reactive protein (CRP) and albumin levels; neutrophil, lymphocyte, and white blood cell (WBC) counts; and the neutrophil/lymphocyte (N/L) ratio all have predictive value in some malignancies [35,36,37]. Another limitation, due to the retrospective design of our study, is the unavailability of FFPE tissue specimens from the primary tumors or brain tissue from resected MBM. Genomic, pharmacogenomic, or PD-L1 expression analysis that could correlate with survival could not be performed. It is suggested that melanoma patients with high immunogenicity (including a high content of immune infiltrating cells, high expression of PD-L1, and high concentration of immune cytokines) and high tumor mutation burden are at a low risk and can benefit more from immunotherapy to achieve prolonged survival [38]. Expression of the PD-1 ligand (PD-L1) was demonstrated in human specimens of melanoma brain metastases [2] and correlated with a higher density of tumor-infiltrating lymphocytes expressing PD-1, suggesting that upregulation of immune checkpoints may explain the ability of brain metastases to bypass the immune system and promote immunosuppression [39]. Drug targeting and pharmacogenomic signatures are of predictive value and, because of the results of our analysis, could separate pts within each mol-GPA prognostic groups into those who are predicted to respond to evaluated treatment option and those who are not. Mutational signatures are markers of the drug sensitivity of cancer cells [40]. This is a good subject for our further investigations in the future.

The concordance of mol-GPA prognosis with modern genetic-driven prognostic models is unknown. There are models for early-stage diseases such as the skin melanoma prognostic signature, which divides patients into three clusters: immunity-high, -medium, and -low. This nomogram model combines the risk scores obtained by the five prognostic genes and other clinicopathological characteristics. It can distinguish and predict skin melanoma patients’ prognosis but was not evaluated in patients with BM [38].

The clinical application of mutational signatures is also supported by the findings dividing cutaneous melanomas (CM) into two groups with distinct clinic, genomic, and functional characteristics: UV-high and UV-low clusters [41]. The UV-low cluster was associated with a low mutational burden and worse overall survival than the UV-high cluster. Those clusters can be distinguished using the panel sequencing data applied in routine clinical practice and could be helpful in the management and predicting prognosis but were not investigated in patients with MBM nor discussed in the context of mol-GPA.

The results of our analysis support the idea of the mol-GPA index being a prognostic not a predictive tool, encouraging the use of all needed treatment interventions regardless of the mol-GPA prognostic group. While the treatment landscape of melanoma is changing, adjuvant and neoadjuvant therapies have become widely used, it would be interesting to observe further development of mol-GPA, especially in the context of secondary CNS dissemination and genetic, genomic, and pharmacogenomic resistance mechanisms.

## 5. Conclusions

Our data show the impact of the treatment modalities on survival before the era of combination immunotherapy with anti-PD1 and anti-CTLA4, as this treatment was not available in our countries in the timeframe included in our analysis. The prognosis of patients with MBM has been improving over the years due to the introduction of the modern treatment options available for patients from all mol-GPA prognostic groups. We have observed significant improvement in survival in the poorest mol-GPA prognostic group, which mandates the use of novel therapies in these patients. In our analysis, the presence of the symptoms associated with brain metastases predicts a worse response to immune checkpoint inhibitors; however, symptoms without steroid use are of less prognostic significance according to our data.

A high frequency of follow-up and the performance of regular MR scans of CNS are justified in the course of melanoma treatment. Close follow-up influences the disease burden at MBM diagnosis. The disease burden is associated with the prognostic factors included in the mol-GPA index, the presence of neurological symptoms and steroid use, and therefore has an influence on patient prognosis. With close follow-up, more asymptomatic patients with a smaller MBM burden and in a better performance status not using glucocorticoids can be diagnosed and treated accordingly, leading to better survival rates.

## Figures and Tables

**Figure 1 cancers-14-05763-f001:**
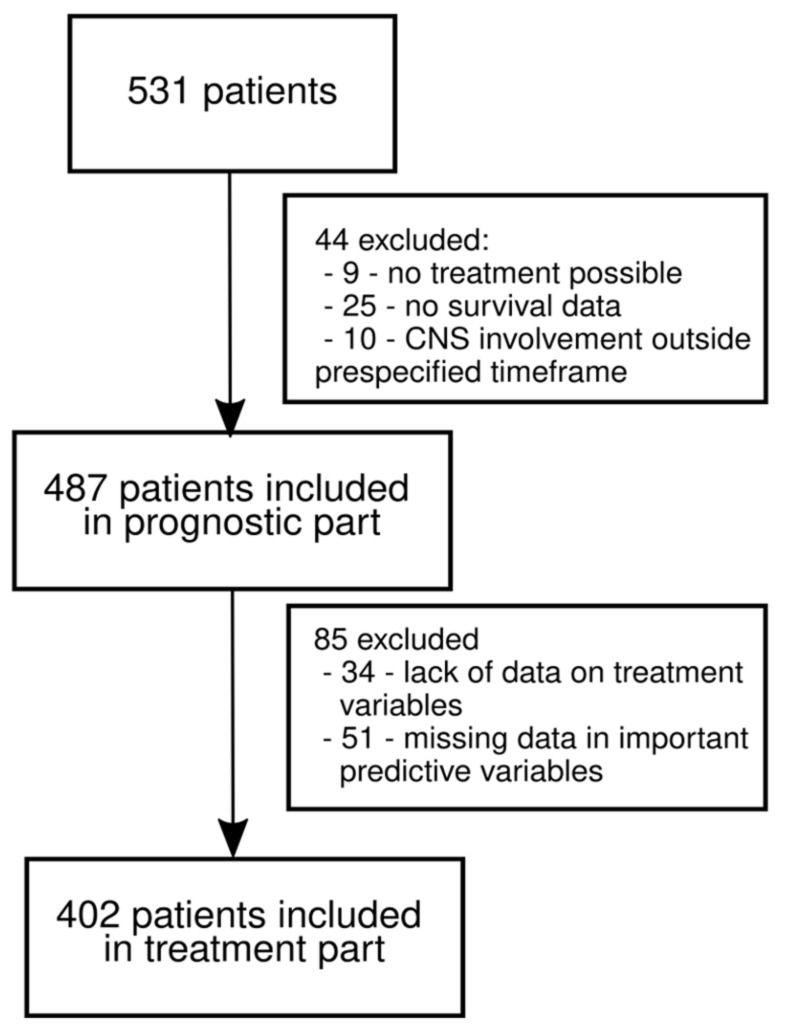
CONSORT plot.

**Figure 2 cancers-14-05763-f002:**
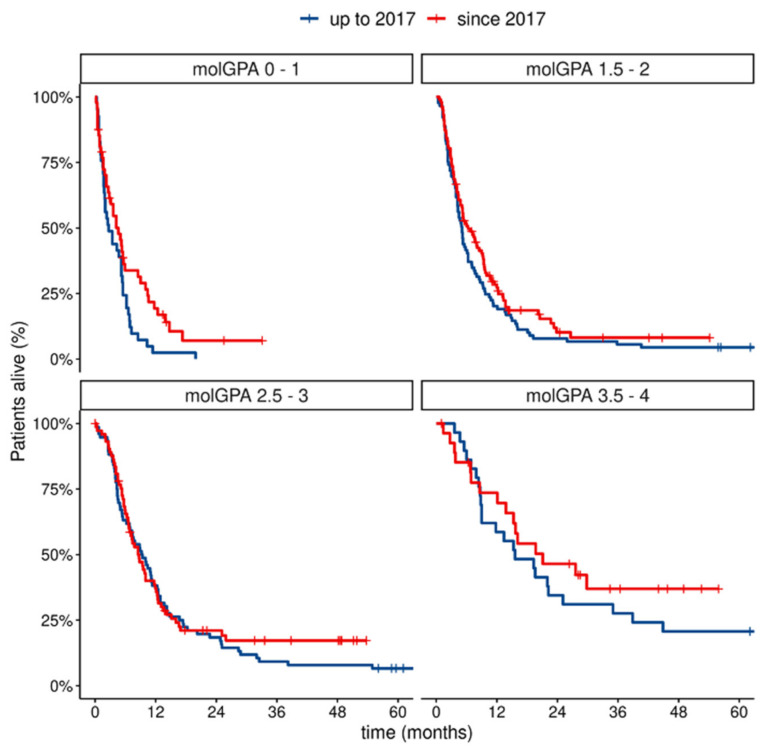
Overall survival by mol-GPA groups and year of MBM diagnosis.

**Figure 3 cancers-14-05763-f003:**
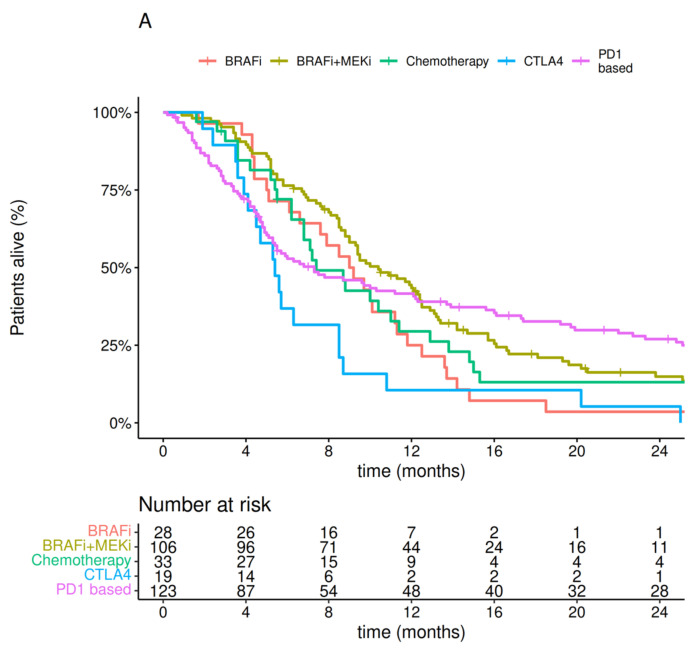
Kaplan–Meier curves for: (**A**) overall survival, (**B**) progression-free survival, and (**C**) intracranial progression-free survival depending on the systemic treatment used in first line after melanoma brain metastases were diagnosed.

**Figure 4 cancers-14-05763-f004:**
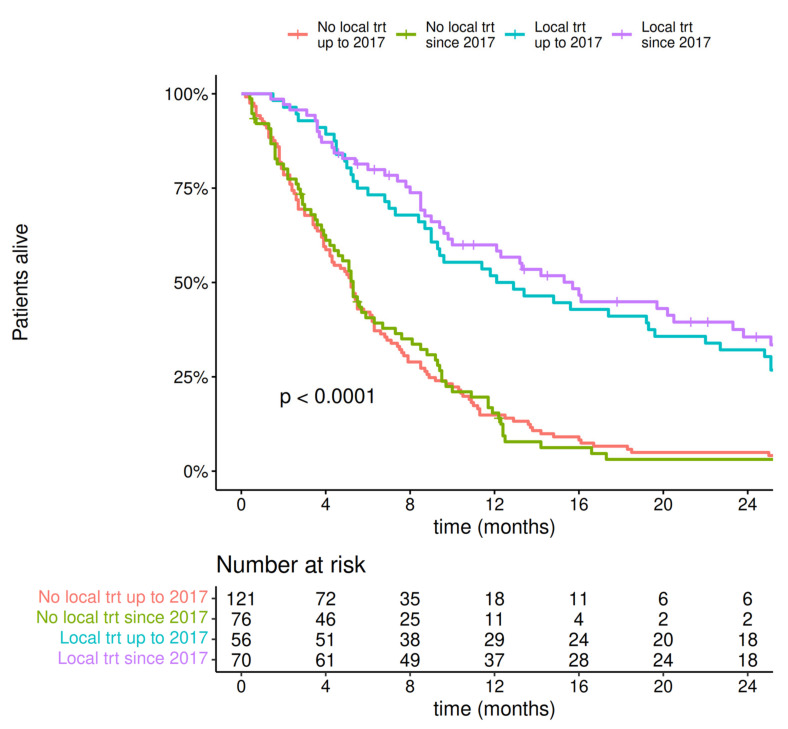
Overall survival of patients depending on implementation of local therapy due to melanoma brain metastases.

**Figure 5 cancers-14-05763-f005:**
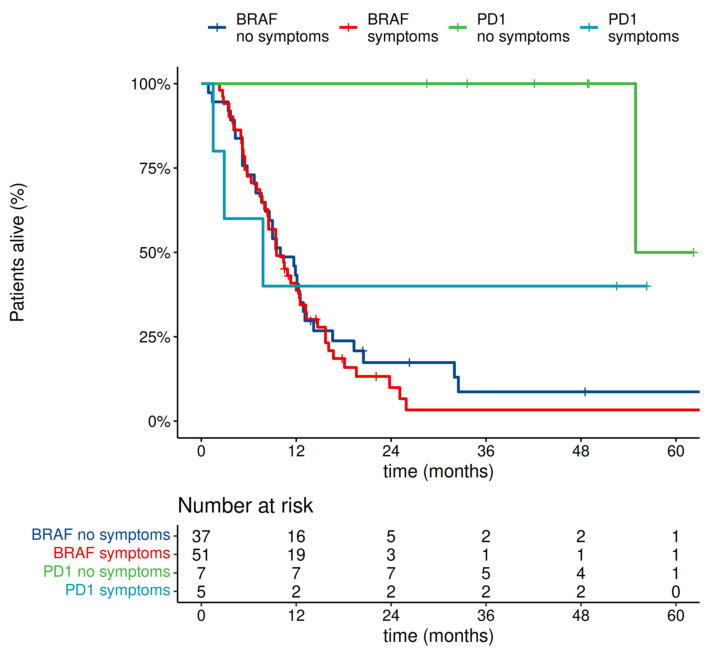
Overall survival in patients with *BRAF* mutation starting systemic therapy due to melanoma brain metastases.

**Table 1 cancers-14-05763-t001:** Patients’ clinical characteristics.

		Prognostic Part		Treatment Part	
Data Analyzed	Variable	Overall	Missing	Overall	Missing
		number of pts (%)	%	number of pts (%)	%
		487		402	
Sex	female	204 (41.9)	0	169 (42.0)	0
	male	283 (58.1)		233 (58.0)	
CNS involvement type	both	14 (2.9)	0	9 (2.2)	0
	brain	468 (96.1)		391 (97.3)	
	meninges	5 (1.0)		2 (0.5)	
*BRAF* mutation status	v600	316 (64.9)	0	255 (63.4)	0
	wt	171 (35.1)		147 (36.6)	
Presence of lung metastases	no	217 (44.6)	0	175 (43.5)	0
	yes	270 (55.4)		227 (56.5)	
Presence of visceral metastases	no	270 (55.4)	0	231 (57.5)	0
	yes	217 (44.6)		171 (42.5)	
No. of MBM	1	131 (26.9)	0	95 (23.6)	0
	2	58 (11.9)		50 (12.4)	
	3	44 (9.0)		39 (9.7)	
	4	15 (3.1)		11 (2.7)	
	5+	239 (49.1)		207 (51.5)	
Diameter of MBM (mm) (median (IQR))		16.00 [9.00, 27.00]	28.1	16.00 [10.00, 27.00]	25.9
Karnofsky performance status	<70	119 (24.4)	0	52 (12.9)	0
	100	53 (10.9)		43 (10.7)	
	70	84 (17.2)		83 (20.6)	
	80	98 (20.1)		95 (23.6)	
	90	133 (27.3)		129 (32.1)	
Presence of CNS symptoms	no	177 (40.7)	10.7	163 (42.7)	5
	yes	258 (59.3)		219 (57.3)	
Use of GCS	no	110 (22.6)	0	106 (26.4)	0
	unknown	102 (20.9)		63 (15.7)	
	yes	275 (56.5)		233 (58.0)	
Use of previous treatment	0	306 (62.8)	0	252 (62.7)	0
	1	128 (26.3)		102 (25.4)	
	2+	53 (10.9)		48 (11.9)	
Extracranial involvement	no	237 (48.7)	0	183 (45.5)	0
	yes	250 (51.3)		219 (54.5)	
Age (median (IQR))		56.00 [44.00, 65.00]	0	56.00 [44.00, 65.00]	0
Previous chemotherapy	no	451 (92.6)	0	368 (91.5)	0
	yes	36 (7.4)		34 (8.5)	
Previous BRAF/MEK inhibitors	no	451 (92.6)	0	372 (92.5)	0
	yes	36 (7.4)		30 (7.5)	
Previous BRAF inhibitors	no	453 (93.0)	0	372 (92.5)	0
	yes	34 (7.0)		30 (7.5)	
Previous ipilimumab	no	460 (94.5)	0	377 (93.8)	0
	yes	27 (5.5)		25 (6.2)	
Previous anti-pd1 antibodies	no	460 (94.5)	0	377 (93.8)	0
	yes	27 (5.5)		25 (6.2)	
No. of pts with MBM diagnosis by year	<2017	235 (48.3)	0	207 (51.5)	0
	≥2017	252 (51.7)		195 (48.5)	

**Table 2 cancers-14-05763-t002:** Two multivariate Cox models pooled from 40 multiple imputed datasets. “Full model” comprises all baseline variables of interest. In the “reduced model,” only variables with *p* < 0.1 in the full model were included.

Analyzed Variables	Full Model				Reduced Model			
Term	HR	Lower 95	Upper 95	*p*-Value	HR	Lower 95	Upper 95	*p*-Value
Sex male	1.16	0.95	1.42	0.139				
Age (per 1 year change)	1.01	1	1.02	0.0056	1.01	1	1.02	0.0048
*BRAF* status (WT vs. mutated)	1.31	1.05	1.64	0.0159	1.33	1.07	1.66	0.0114
Lung metastases (yes vs. no)	1.24	1.01	1.54	0.0446	1.25	1.01	1.54	0.0375
Visceral metastases (yes vs. no)	1.3	1.05	1.61	0.0168	1.27	1.03	1.57	0.0239
Other extracranial metastases (yes vs. no)	0.98	0.8	1.21	0.8665				
No. of CNS metastases = 1	Reference				Reference			
No. of CNS metastases = 2–4	1.65	1.23	2.21	0.0009	1.6	1.19	2.13	0.0017
No. of CNS metastases >5	2.1	1.61	2.73	<0.0001	2.08	1.61	2.7	<0.0001
Diameter of biggest CNS meta (per 1mm change)	1	0.99	1.01	0.3625				
Year of MBM diagnosis (per 1 year change since 2014)	0.91	0.85	0.97	0.0038	0.91	0.86	0.97	0.0045
Karnofsky performance status score (per 10 pts change)	0.9	0.83	0.98	0.011	0.9	0.83	0.97	0.009
No CNS symptoms, no GCS use	Reference				Reference			
No CNS symptoms, GCS due to other causes	1.42	1	2.03	0.0525	1.42	0.99	2.02	0.0549
CNS symptoms, no GKS use	1.16	0.8	1.69	0.4309	1.22	0.85	1.76	0.2832
CNS symptoms requiring GCS	1.63	1.23	2.16	0.0007	1.69	1.29	2.21	0.0001
No previous systemic treatment	Reference				Reference			
1 line of previous systemic treatment	1.65	1.31	2.08	<0.0001	1.65	1.31	2.09	<0.0001
>1 line of previous systemic treatment	1.62	1.16	2.27	0.0051	1.57	1.14	2.16	0.0062

**Table 3 cancers-14-05763-t003:** Summary of hazard ratio (HR) for OS by mol-GPA groups.

Term	HR	Lower_95	Upper_95	*p*-Value
molGPA 3.5–4	Reference			
molGPA 2.5–3	1.88	1.31	2.7	0.0007
molGPA 1.5–2	2.83	1.98	4.03	<0.0001
molGPA 0–1	4.84	3.26	7.17	<0.0001

**Table 4 cancers-14-05763-t004:** HR according to glucocorticoids (GCs) usage and symptoms.

Factor	HR	*p*-Value	Lower 95	Upper 95
for no symptoms and no GCs as a reference group				
pts with no symptoms and on GCS	1.81	0.0006	1.29	2.53
pts with symptoms without GCS	1.26	0.1767	0.90	1.78
pts with symptoms on GCS	1.98	0.0000	1.56	2.53

## Data Availability

Data supporting the reported results can be retrieved from P.T.: pawel.teterycz@pib-nio.pl. They have archived the datasets analyzed or generated during the study. All data are available for research cooperation purposes upon Data Transfer Agreement (DTA) approval.

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
