# Peer review of "The Analysis of Trends in Survival for Patients with Melanoma Brain Metastases with Introduction of Novel Therapeutic Options before the Era of Combined Immunotherapy—Multicenter Italian–Polish Report"

_cancers, 2022, doi:10.3390/cancers14235763_

Round 1
Reviewer 1 Report
The authors analyzed trends in survival for patients with melanoma 3 brain metastases with introduction of novel therapeutic options 4 before the era of combined immunotherapy. The investigation is so interesting, however, I have some concerns to be discussed.
・Can the mol-GPA index be applied as an indicator of metastases other than brain metastases?
・How should the mol-GPA index be improved in the future?
・Is there a difference in prognosis depending on the presence or absence of symptoms of brain metastases?
・Does inflammation play a role in the mol-GPA prognostic group?
Please refer to the following literature for discussion
-Characterizing inflammatory markers in highly aggressive soft tissue sarcomas. Medicine, 101(39), e30688. https://doi.org/10.1097/MD.0000000000030688
Author Response
Response to Reviewer 1 Comments
Point 1: Can the mol-GPA index be applied as an indicator of metastases other than brain metastases?
Response 1: Mol- GPA index, created specifically for the patients with melanoma brain metastases incorporates such factors as KPS, age, number of brain metastases, BRAF status and coexistence of extracranial metastases. In melanoma BRAF mutational status is closely related to more aggressive behavior of melanoma in metastatic setting and predictor for the choice of therapy. There are other disease specific GPA indexes all of them dedicated to prediction of survival for patients with brain metastases. Survival and factors that predict survival, according to the definition, vary by diagnosis and the molecular profile of the patient’s tumor histopathology. There are other GPA; disease specific, organ specific GPA indices. Unique prognostic models were developed for patients with non-small cell lung cancer, small cell lung cancer, melanoma, renal cell carcinoma, breast cancer and gastrointestinal cancer. Organ specific factors that have significant influence on prognosis were grouped into prognostic indices, graded scales as Diagnosis-Specific Graded Prognostic Assessment (DS-GPA).[1]
Point 2: How should the mol-GPA index be improved in the future?
Response 2: The influence of neurological symptoms and steroid usage at diagnosis and before treatment of BM could be separately evaluated and if important, separately added to other factors[2], KPS is not specific, we don’t know if in this particular patient it deals with intra- or extracranial symptoms, if there is KPS80 and extracranial disease in M1d patient we do not know if it is from BM or other metastases. As 50% of melanoma M1 patients develop BM, many of them had previous treatment lines e.g., in adjuvant setting. Systemic treatment in the adjuvant setting before MBM development may influence prognosis- this is an interesting group of pts to be investigated in the context of mol-GPA. Melanoma mol-GPA does not include information about previous treatment lines when brain dissemination is a secondary diagnosis. It could be investigated. There will be more secondary MBM evolved for example after adjuvant treatment- this is a new situation for mol-GPA development. There might be different prognosis due to secondary resistance mechanisms.
Thank you for your comment- we have incorporated our answer into the manuscript.
Point 3: Is there a difference in prognosis depending on the presence or absence of symptoms of brain metastases?
Response 3: According to our findings symptoms do not interfere with prognosis. This thesis is supported by other retrospective findings, where presenting symptoms were found of none influence on prognosis[3] Prospective evaluation would be of great value.
Symptoms might be associated with higher disease burden or rapid disease growth deteriorating survival prognosis. Symptoms may relate to the localization of MBM in the brain – metastases in eloquent regions (as in the left temporal and frontal lobes for speech and language, bilateral occipital lobes for vision, bilateral parietal lobes for sensation, and bilateral motor cortex for movement) might even have a positive influence on prognosis leading to earlier diagnosis and local therapies being quickly applied.
That is why we would like to emphasize the need of distinction neurological symptoms at the time of systemic treatment introduction (related to the disease burden and requiring steroids) from the symptoms present at diagnosis, which resolve by the treatment start and do not require steroids at the treatment start. The use of glucocorticosteroids seems to worsen prognosis, not the presence of symptoms at diagnosis.
There are clinical trials (CheckMate 204, Australian ABC study) evaluating an influence of ICI (immune checkpoint inhibitors) treatment on survival of pts with MBM. The use of steroids from presence of steroid-free symptoms was not distinguished in those trials and evaluated in the same cohorts. In ABC study symptomatic pts were in one cohort with those with leptomeningeal spread. It is unclear if lack of response in symptomatic cohorts is due to a higher tumor burden and rapid growth, to more immune-resistant phenotype, low distribution of tumor-infiltrating lymphocytes, or to immune suppression induced by corticosteroids.[2]
At this point, if we deprive symptomatic pts of the immunotherapy based on the results of these trials we do influence their prognosis, because some of them can achieve durable responses.[2,4]
Thank you for to this question- our response was incorporated into the manuscript.
Point 4: Does inflammation play a role in the mol-GPA prognostic group?
Please refer to the following literature for discussion
-Characterizing inflammatory markers in highly aggressive soft tissue sarcomas. Medicine, 101(39), e30688. https://doi.org/10.1097/MD.0000000000030688
Response 4: Inflammation and prognosis seem to be very interesting to investigate. Measuring pretreatment inflammatory peripheral blood markers has a predictive value in some malignancies[5–7] Investigated inflammatory peripheral blood markers are: CRP C-reactive protein (CRP) and albumin levels; neutrophil, lymphocyte, and white blood cell (WBC) counts; and neutrophil/lymphocyte (N/L) ratio and they may impact prognosis of pts with MBM.
In our study mol-GPA is to be evaluated at diagnosis of MBM – in most cases there are no blood samples to be evaluated due to the retrospective character of our analysis. No comparable and standardized blood analysis would be possible to obtain- data would be missing for appropriate statistical analysis.
It is a very interesting field to be investigated in our future prospective research addressing the subject of possible association among mol-GPA index factors, pretreatment inflammatory peripheral blood markers and patients prognosis, as was nicely shown in highly aggressive soft tissue sarcomas[6]
This remark made us think more about our future plans and we have incorporated our response into the manuscript.
- Sperduto, P.W.; Kased, N.; Roberge, D.; Xu, Z.; Shanley, R.; Luo, X.; Sneed, P.K.; Chao, S.T.; Weil, R.J.; Suh, J.; et al. Summary Report on the Graded Prognostic Assessment: An Accurate and Facile Diagnosis-Specific Tool to Estimate Survival for Patients with Brain Metastases. J Clin Oncol 2012, 30, 419–425, doi:10.1200/JCO.2011.38.0527.
- Tawbi, H.A.; Forsyth, P.A.; Hodi, F.S.; Lao, C.D.; Moschos, S.J.; Hamid, O.; Atkins, M.B.; Lewis, K.; Thomas, R.P.; Glaspy, J.A.; et al. Safety and Efficacy of the Combination of Nivolumab plus Ipilimumab in Patients with Melanoma and Asymptomatic or Symptomatic Brain Metastases (CheckMate 204). Neuro-Oncol. 2021, 23, 1961–1973, doi:10.1093/neuonc/noab094.
- Bander, E.D.; Yuan, M.; Carnevale, J.A.; Reiner, A.S.; Panageas, K.S.; Postow, M.A.; Tabar, V.; Moss, N.S. Melanoma Brain Metastasis Presentation, Treatment, and Outcomes in the Age of Targeted and Immunotherapies. Cancer 2021, 127, 2062–2073, doi:10.1002/cncr.33459.
- Long, G.V.; Atkinson, V.; Lo, S.; Guminski, A.D.; Sandhu, S.K.; Brown, M.P.; Gonzalez, M.; Scolyer, R.A.; Emmett, L.; McArthur, G.A.; et al. Five-Year Overall Survival from the Anti-PD1 Brain Collaboration (ABC Study): Randomized Phase 2 Study of Nivolumab (Nivo) or Nivo+ipilimumab (Ipi) in Patients (Pts) with Melanoma Brain Metastases (Mets). J. Clin. Oncol. 2021, 39, 9508–9508, doi:10.1200/JCO.2021.39.15_suppl.9508.
- Proctor, M.J.; Morrison, D.S.; Talwar, D.; Balmer, S.M.; O’Reilly, D.S.J.; Foulis, A.K.; Horgan, P.G.; McMillan, D.C. An Inflammation-Based Prognostic Score (MGPS) Predicts Cancer Survival Independent of Tumour Site: A Glasgow Inflammation Outcome Study. Br. J. Cancer2011, 104, 726–734, doi:10.1038/sj.bjc.6606087.
- Hashimoto, K.; Nishimura, S.; Shinyashiki, Y.; Ito, T.; Akagi, M. Characterizing Inflammatory Markers in Highly Aggressive Soft Tissue Sarcomas. Medicine (Baltimore) 2022, 101, e30688, doi:10.1097/MD.0000000000030688.
- Pal, S.; Nath, S.; Meininger, C.J.; Gashev, A.A. Emerging Roles of Mast Cells in the Regulation of Lymphatic Immuno-Physiology. Front. Immunol. 2020, 11, 1234, doi:10.3389/fimmu.2020.01234.

Reviewer 2 Report
This is an interesting study, and the authors retrospectively analyzed and found that significant improvement in survival in the mol-GPA poorest prognostic group in melanoma patients. In their analysis the presence of symptoms associated with brain metastases predicts worse response to anti-PD-1 or anti-CTLA4. In sum, they showed that systemic therapy improves outcomes when was combined with local therapy and local and systemic treatment significantly prolong survival for poorest mol-GPA prognosis.
Questions:
1. It will raise the value of this manuscript if authors check the expression level of PD-L1 in tumor tissue and whether the expression of PD-L1 correlates with survival. In addition, the authors may check the PD-L1 expression level in different GPA grades.
2. I suggested to the authors to include information about drug target or pharmacogenomic signatures related to mol-GPA in the discussion and discuss whether this mol-GPA not only predict survival but also offer a therapeutic strategy depending on mol-GPA results.
3. Please describe in more detail mol-GPA in the abstract and introduction. And whether this model also can apply to other cancers.
4. So far, the resistance to immunotherapy is the critical question in the clinic. Therefore, could this mol-GPA predict which groups of GPA grade may develop a resistance melanoma?
Author Response
Question 1. It will raise the value of this manuscript if authors check the expression level of PD-L1 in tumor tissue and whether the expression of PD-L1 correlates with survival. In addition, the authors may check the PD-L1 expression level in different GPA grades.
Response 1: The limitation of our study, due to its retrospective design, is lack of FFPE tissue specimens from primary tumors or brain tissue from resected MBM allowing PD-L1 expression analysis, that could correlate with survival. As it is shown by current studies suggesting that melanoma patients with high immunogenicity (including high content of immune infiltrating cells, high expression of PD-L1 and high concentration of immune cytokines) and high tumor mutation burden are at low risk and can benefit more from immunotherapy achieving prolonged survival.[1]
Expression of the PD-1 ligand (PD-L1) was demonstrated in human specimens of brain metastases from melanoma[2] and correlated with higher density of tumor-infiltrating lymphocytes expressing PD-1, suggesting that upregulation of immune checkpoints may be important for the ability of brain metastases to evade the immune system and enhance immunosuppression.[2]
Thank you for your suggestions, the response was incorporated into the manuscript, we plan to collect the tissue for further analyses in the frame of scientific grant.
Question 2. I suggested to the authors to include information about drug target or pharmacogenomic signatures related to mol-GPA in the discussion and discuss whether this mol-GPA not only predict survival but also offer a therapeutic strategy depending on mol-GPA results.
Response 2: Drug targeting and pharmacogenomic signatures might have an importance due to the profound influence of ICI treatment on pts survival nowadays. Mutational signatures are markers of drug sensitivity of cancer cells.[3]
Drug targeting and pharmacogenomic signatures are of predictive value, and because of the results of our analysis – could distinguish pts within each mol-GPA prognostic groups into those who are predicted to respond to evaluated treatment option or who are not. However, our treatment groups are too small to conduct statistically significant analysis at this point. It could be the subject of our further investigations in the future.
Concordance of mol-GPA prognosis with modern genetic-driven prognostic models is unknown.
There are models: for early- stage disease as skin melanoma prognostic signature that divides patients into three clusters: [1] immunity-high, -medium and -low. This nomogram model combines the risk scores obtained by the five prognostic genes and other clinicopathological characteristics. It can distinguish and predict skin melanoma patients’ prognosis but was not evaluated for pts with BM.
Clinical application of mutational signatures is supported also by the findings dividing cutaneous melanomas (CM) into two groups with distinct clinic-genomic and functional characteristics: UV-high and UV-low clusters.[4] CMs belonging to the UV-low cluster were associated with a low mutational burden and worse overall survival than the UV-high cluster. UV-high and UV-low clusters can be distinguished using panel sequencing data, in routine clinical practice. This clustering is useful for management and for making prognosis but was not investigated in pts with MBM.
The results of our analysis support the idea of mol-GPA index being a prognostic not predictive tool encouraging the use of all needed treatment interventions regardless of mol-GPA prognostic group.
Thank you for the remark, our response was incorporated into the manuscript.
Question 3. Please describe in more detail mol-GPA in the abstract and introduction. And whether this model also can apply to other cancers.
Response 3: Thank you for that suggestion - we have followed it and enriched our manuscript with the information:
The prognosis of patients with MBM can be estimated and depends on many factors and among them are histology and molecular profile of a tumor. Organ specific factors that have significant influence on prognosis were grouped into prognostic indices, graded scales as Diagnosis-Specific Graded Prognostic Assessment (DS-GPA).[5] Unique prognostic models were developed for patients with non-small cell lung cancer, small cell lung cancer, melanoma, renal cell carcinoma, breast cancer and gastrointestinal cancer. Melanoma mol-GPA is composed of 5 factors: age, Karnofsky Performance Score (KPS), extracranial metastases, number of brain metastases, BRAF status. Factors score a 0, 0.5, or 1.0 value. From these data, the GPA score is calculated and specific survival predicted. The best prognosis has mol-GPA group of 3.5- of 4.0 points where OS reaches 34 months, while in mol-GPA 0–1-point group it is only 5 months. GPA is designed to distinguish classes of patients by prognosis before treatment. Additionally, in melanoma BRAF mutational status is closely related to more aggressive behavior of melanoma in metastatic setting and predictor for the choice of therapy.
Unfortunately, abstract cannot be expanded due to 200 words limitation.
Question 4. So far, the resistance to immunotherapy is the critical question in the clinic. Therefore, could this mol-GPA predict which groups of GPA grade may develop a resistance melanoma?
Response 4: In our analysis we wanted to emphasize that GPA is a prognostic, not predictive marker, as with the assumptions of its creators.[5]Our results have shown, that patients with BM benefit when all modern therapeutic approaches are incorporated into their treatment plan regardless of mol-GPA prognostic group.
- Tian, Q.; Gao, H.; Zhao, W.; Zhou, Y.; Yang, J. Development and Validation of an Immune Gene Set-Based Prognostic Signature in Cutaneous Melanoma. Future Oncol. 2021, 17, 4115–4129, doi:10.2217/fon-2021-0104.
- Berghoff, A.S.; Ricken, G.; Widhalm, G.; Rajky, O.; Dieckmann, K.; Birner, P.; Bartsch, R.; Höller, C.; Preusser, M. Tumour-Infiltrating Lymphocytes and Expression of Programmed Death Ligand 1 (PD-L1) in Melanoma Brain Metastases. Histopathology 2015, 66, 289–299, doi:10.1111/his.12537.
- Levatić, J.; Salvadores, M.; Fuster-Tormo, F.; Supek, F. Mutational Signatures Are Markers of Drug Sensitivity of Cancer Cells. Nat. Commun. 2022, 13, 2926, doi:10.1038/s41467-022-30582-3.
- Kim, Y.-S.; Lee, M.; Chung, Y.-J. Two Subtypes of Cutaneous Melanoma with Distinct Mutational Signatures and Clinico-Genomic Characteristics. Front. Genet. 2022, 13.
- Sperduto, P.W.; Kased, N.; Roberge, D.; Xu, Z.; Shanley, R.; Luo, X.; Sneed, P.K.; Chao, S.T.; Weil, R.J.; Suh, J.; et al. Summary Report on the Graded Prognostic Assessment: An Accurate and Facile Diagnosis-Specific Tool to Estimate Survival for Patients with Brain Metastases. J Clin Oncol 2012, 30, 419–425, doi:10.1200/JCO.2011.38.0527.

Round 2
Reviewer 1 Report
The authors replied well, so the manuscript is suitable for publication.
Reviewer 2 Report
I am satisfied with the author's responses to the questions raised in my initial review.